# Musical Experience Relates to Insula-Based Functional Connectivity in Older Adults

**DOI:** 10.3390/brainsci12111577

**Published:** 2022-11-19

**Authors:** Meishan Ai, Psyche Loui, Timothy P. Morris, Laura Chaddock-Heyman, Charles H. Hillman, Edward McAuley, Arthur F. Kramer

**Affiliations:** 1Department of Psychology, Northeastern University, Boston, MA 02115, USA; 2Department of Music, Northeastern University, Boston, MA 02115, USA; 3Department of Physical Therapy, Movement & Rehabilitation Sciences, Northeastern University, Boston, MA 02115, USA; 4Beckman Institute, University of Illinois at Urbana-Champaign, Urbana, IL 61801, USA; 5Department of Kinesiology and Community Health, University of Illinois at Urbana-Champaign, Urbana, IL 61801, USA

**Keywords:** music, functional connectivity, insula, older adults

## Abstract

Engaging in musical activities throughout the lifespan may protect against age-related cognitive decline and modify structural and functional connectivity in the brain. Prior research suggests that musical experience modulates brain regions that integrate different modalities of sensory information, such as the insula. Most of this research has been performed in individuals classified as professional musicians; however, general musical experiences across the lifespan may also confer beneficial effects on brain health in older adults. The current study investigated whether general musical experience, characterized using the Goldsmith Music Sophistication Index (Gold-MSI), was associated with functional connectivity in older adults (age = 65.7 ± 4.4, *n* = 69). We tested whether Gold-MSI was associated with individual differences in the functional connectivity of three a priori hypothesis-defined seed regions in the insula (i.e., dorsal anterior, ventral anterior, and posterior insula). We found that older adults with more musical experience showed greater functional connectivity between the dorsal anterior insula and the precentral and postcentral gyrus, and between the ventral anterior insula and diverse brain regions, including the insula and prefrontal cortex, and decreased functional connectivity between the ventral anterior insula and thalamus (voxel *p* < 0.01, cluster FWE *p* < 0.05). Follow-up correlation analyses showed that the singing ability subscale score was key in driving the association between functional connectivity differences and musical experience. Overall, our findings suggest that musical experience, even among non-professional musicians, is related to functional brain reorganization in older adults.

## 1. Introduction

Individuals show different trajectories in cognitive health during adult aging, with some people maintaining cognitive function well into old age while others show decline [1,2]. Recent years have seen increasing interest in physically and mentally stimulating activities that seemed to protect older adults from cognitive decline and neuropathology [3,4]. As a complex cognitive task that integrates multisensory inputs and requires complex motor outputs, musical activity, such as singing or playing an instrument, involves sensory, motor, and cognitive systems [5]. Thus, musical activity can be seen as one type of mental stimulation that might enhance cognitive health in older adults. Several reports have shown direct benefits of deliberate musical practice in auditory and motor functions among older adults [6,7,8]. While there is evidence that maintained sensory functions strongly link to better cognitive function in later life [9,10], the potential long-transfer benefits from music to higher level cognition is still mixed. A meta-analysis of transfer effects of musical experience in aging have shown effects of musical experience on domain-general cognitive functions such as processing speed, attention, and episodic memory [11]. However, the modulation of musical experience was not consistently observed across all cognitive domains [12,13], which indicates a domain-specific effect. Moreover, a meta-analysis found a smaller effect on cognition for music training with higher quality of study design [14]. Given these mixed findings, it is important to investigate the neurobiological or physiological mechanisms behind the musical experience, to further disentangle the relationship between musical experience and behavioral function in later life. While these effects were studied with some electrophysiological measures (e.g., brainstem coding of speech) [15], less is known about the extent to which lifetime musical experience in non-musicians alters structural and functional measures of brain connectivity, as quantified by magnetic resonance imaging (MRI) in older adults.

Neuroimaging evidence revealed greater grey matter volume and functional connectivity within and between auditory and motor systems for musicians compared to non-musicians, or musicians with more music experience compared to those with less experience [16,17,18]. Additionally, the effects of musical practice extend beyond unimodal brain regions such as auditory or motor cortices. Musical experiences, such as playing a musical instrument, require the brain’s ability to combine information from multiple sensory resources. As such, musical practice also involves multimodal processing, thus recruiting brain regions that integrate multiple types of sensory information [19]. The insula is one of the regions that integrates large scale sensory, motor, cognitive and affective processing networks [20,21], and is hypothesized to be a key overlapping region between auditory prediction and reward networks for music processing [22]. As a core node of the salience network, the insula aids in the detection of salient events and interacts with other large scale networks, thus facilitating the identification of internal and external stimuli [23]. The insula is anatomically and functionally segregated into dissociable zones. Information integration in the insula is thought to be self-organized in a posterior-to-anterior sequence, i.e., the posterior insula integrates primary sensory information from the body while the anterior insula represents subjective evaluation of these inputs [24,25]. Similarly, a meta-analysis has summarized that the anterior parts of the insula are more specific to socio-emotional and cognitive function, while middle and posterior insula are related to olfactory–gustatory and sensorimotor systems [21]. Deen et al. (2011) revealed three sub-regions of the insula using cluster analysis with distinct functional networks based on resting-state functional imaging data: dorsal anterior insula, ventral anterior insula, and posterior insula, being related to cognitive control, affective processing, and sensorimotor regions, respectively [26]. 

Several recent studies have found that musical experience might modulate the function of the insula. Singers showed altered activation patterns in the right anterior insula compared to non-singers while manipulating auditory feedback, which revealed the experience-dependent modulation in insular function [27,28]. Insula function was also associated with emotion reactions from music [29]. Musical experience also modulates functional connectivity between the insula and other brain regions. Lordier et al. (2019) revealed the effect of exposure to music on connectivity between the salience network and sensorimotor networks, and between the salience network and thalamus and precuneus networks in newborns [30]. Further, a previous study found that musicians showed greater functional connectivity between the insula and a broad range of regions that involve salience detection, executive control and affective processing, by seeding in the dorsal anterior, ventral anterior, and posterior insula regions [31]. 

The insula was also associated with memory decline in older adults [32]. Moreover, it has been found to be a critical region in the early stages of dementia with Lewy bodies [33,34]. As an overlapping region between the auditory and reward systems, the insula is a core of the disrupted brain network in musical anhedonia [22]. This is supported by seed-based functional connectivity from the same auditory and reward systems, which show overlap in the anterior insula; this overlap was significantly decreased in Alzheimer’s disease (AD) patients compared to age-matched individuals with mild cognitive impairment (MCI), who were relatively similar in functional connectivity to cognitively healthy older adults [35]. Furthermore, a recent study found a positive association between insula volume and musical training in the same sample of healthy older adults as used in the current analysis [36]. Together, these findings suggest that the insula may be an important neural mechanism in how musical experience influences behavioral function in later life.

Investigating the effect of lifetime musical experience on insula-based networks in older adults may provide insight into understanding how music shapes information integration behaviors. Here, we used the Gold-MSI to characterize lifetime musical experience in older adults [37]. The Gold-MSI is a simple survey that has been normed in over 141,000 individuals across the human lifespan. It contains questions not only on singing ability and musical listening skills, but also on engagement in musical activities and emotional responses to music, thus allowing us to capture multiple facets of musical behaviors and expertise, in non-musicians [37]. We used this index to capture a musical profile for individuals who were not recruited on the basis of professional musical training, but nonetheless fall along a range of musical experience over the human lifespan. 

The current study examined whether non-professional musical experiences (i.e., musical sophistication) influences functional connectivity in insula-based networks in older adults. We tested the hypothesis that musical experience is associated with functional connectivity of insula-based networks by relating seed-based functional connectivity in resting-state functional MRI (fMRI) from seed regions of dorsal anterior, ventral anterior, and posterior insula to individual differences in musical experience as quantified by the Gold-MSI and its subscales. 

## 2. Materials and Methods

### 2.1. Participants

All participants in the current study were healthy, relatively inactive older adults who had previously participated in a randomized controlled trial of exercise effects on cognition and brain health (https://clinicaltrials.gov/ct2/show/NCT01472744 (accessed on 3 August 2021)). All imaging data and demographic information were taken from the pre-intervention assessments (and therefore not influenced by the exercise intervention). Participants provided written informed consent in accordance with the Institutional Review Board of the University of Illinois at Urbana-Champaign. Inclusion criteria from the original study were: (1) aged from 60 to 79; (2) scored 21 or higher on the Telephone Interview of Cognitive Status (TICS-M) questionnaire; (3) corrected acuity of 20/40 or better for both eyes and no diagnosis of color-blindness; (4) depression score on the Geriatric Depression Scale (GDS-15) lower than 10; (5) right-handed; (6) no history of brain surgery that involved removal of brain tissue; (7) no history of stroke or transient ischemic attack (TIA); (8) being MRI-compatible. 

Participants completed the Gold-MSI after completing a new consent form for this measurement between 5 and 8 years after the pre-intervention imaging data. The Gold-MSI questionnaire and a consent form were mailed to 214 participants, and 73 older adults returned the questionnaire with completed data. The final sample in the current study included 69 participants who had completed both the Gold-MSI questionnaire and the MRI session (age = 65.70 ± 4.37; sex, male = 19, female = 50; years of education = 16.17 ± 2.65). None of the participants were professional musicians but had a varied amount of musical experience (see Figure 1). The median scores of the five subscales and the total are displayed in Figure 1, along with the median of population norm (*n* = 147,633) for comparison.

### 2.2. The Goldsmiths Musical Sophistication Index Questionnaire

The Gold-MSI was administered to assess musical sophistication, which refers to music-related behaviors, skills, and achievements [37]. Besides formal musical training and knowledge, this instrument contains a broad range of facets of musical behavior and experience, including musical skills, engagement, and emotional responses. As musical sophistication is necessarily a multi-faceted domain, relating the overall score as well as subscale scores of the Gold-MSI to brain connectivity patterns allows us to explore individual differences in multiple aspects of musical experiences in the way they relate to individual differences in brain structure and function. The questions in Gold-MSI include five subscales as follows:Active Engagement

This is a 9-item subscale to assess musical engagement behaviors (e.g., “I keep track of new music that I come across”), and the allocation of time and money on musical activities (e.g., “I listen attentively to music for ____ hours per day).

Perceptual Ability

This is a 9-item subscale for self-reported perceptual ability in music, most of which are related to music listening skills (e.g., “I can tell when people sing or play out of tune”).

Musical Training

This is a 7-item subscale asking about the extent of musical training and practice (e.g., “I engaged in regular daily practice of a musical instrument including voice for ____ years”), and the degree of self-assessed musicianship (e.g., “I would not consider myself a musician”).

Emotion

This is a 6-item subscale to assess behaviors related to emotional responses to music (e.g., “I am able to talk about the emotions that a piece of music evokes in me”).

Singing Ability

This is a 7-item subscale for self-reported skills and activities related to singing (e.g., “After hearing a new song two or three times I can usually sing it by myself”).

In the current analysis, we examined the association between the music composite score (i.e., the sum of all five subscale scores) and the insula-based functional connectivity across the whole brain. We then examined the correlations between insula-based functional connectivity and scores on all five subscales, as a follow-up analysis to identify any specific aspects of musical experiences that might influence insula-based functional connectivity more than others.

### 2.3. Imaging Data Acquisition

Participants were scanned using a 3 Tesla Siemens Trio Tim system. High-resolution T1-weight structural data were acquired using a 3D MPRAGE (Magnetization Prepared Rapid Gradient Echo Imaging) sequence. The parameters were: repetition time (TR) = 1900 ms, echo time (TE) = 2.32 ms, inversion time (TI) = 900 ms, flip angle = 9°, FoV = 230 mm, resolution = 0.9 × 0.9 × 0.9 mm, GRAPPA acceleration factor = 2). T2*-weight resting state data were acquired using a fast echo-planar imaging (EPI) sequence. The parameters were: 6 min of duration, TR = 2 s, TE = 25 ms, flip angle = 80°, 3.4 × 3.4 mm^2^ in-plane resolution, 35 4 mm thick slices acquired in ascending order, GRAPPA acceleration factor = 2, 64 × 64 matrix).

### 2.4. Data Analysis

#### 2.4.1. Data Preprocessing

The preprocessing of the resting state functional connectivity data was performed using the default preprocessing pipeline in the CONN-toolbox v. 20c [38], which is based on Statistical Parametric Mapping (SPM) v.12 (https://www.fil.ion.ucl.ac.uk/spm/software/spm12/ (accessed on 27 May 2021)) and MATLAB v. 2020b (The MathWorks, Natick, MA, USA). The processing pipeline includes functional realignment and unwarping, slice-timing correction, outlier identification, functional smoothing, segmentation and normalization. Images were segmented into grey matter, white matter, and CSF tissue, and normalized into Montreal Neurological Institute (MNI) space. Finally, all images were smoothed using a 6 mm Gaussian kernel. The outlier identification flagged acquisition with framewise displacement above 0.9 mm or global BOLD signal changes above 5 s.d. thresholds as potential outliers. The default denoising pipeline in CONN included linear regression of potential confounding effects and temporal band-pass filtering. An anatomical component-based noise correction procedure (aCompCor) was implemented for reduction of noise [39], and the noise components included cerebral white matter and cerebrospinal areas, estimated subject-motion parameters, scrubbing, and constant and first-order linear session effects. One participant was removed from the final analysis because of observation of potential clinical neuropathology in brain structure. Two other participants were removed from the final analysis because of having more than 40 scans flagged as outliers. This criteria was decided based on having at least 5 min of scanning time for resting state [40].

#### 2.4.2. Seed-to-Voxel Analysis

Three a priori regions of interests (ROIs) within the insula were defined based on a previous cluster analysis, which divided the insula into dorsal anterior, ventral anterior and posterior regions [26], shown in Figure 2. The mean time-series within each ROI were computed and correlated with the time-series in each voxel of the whole brain and converted to normally distributed z-scores using Fisher transformation. The effect of general Gold-MSI score, controlling for age, sex, educational level and mean motion included as covariates was assessed in a second-level general linear model, and significant clusters were determined using a height-level statistical threshold of *p* < 0.01 and a cluster threshold of *p* < 0.05 family-wise effort (FWE)-corrected. 

#### 2.4.3. Behavioral Data Correlation Analysis

As a follow-up analysis to see which specific facets of musical experience were associated with functional connectivity, we conducted pairwise correlations between the above Fisher-transformed z-scores of the Pearson correlation coefficient between the clusters and seeds, and the Gold-MSI subscale scores of Active Engagement, Perceptual Ability, Musical Training, Emotion, and Singing Ability.

## 3. Results

### 3.1. The Association between Musical Experience and Insula Seed Functional Connectivity

The seed-to-voxel analysis revealed that general Gold-MSI scores were positively associated with insula seed-based functional connectivity. When seeding the bilateral dorsal anterior insula, participants with higher Gold-MSI scores showed greater functional connectivity between this region and a significant cluster in the precentral and postcentral gyri (Figure 3a). When seeding the bilateral ventral anterior insula, higher Gold-MSI scores were positively associated with functional connectivity between this region and the right insula (Figure 3b) and right inferior frontal gyrus (Figure 3b), and negatively associated with functional connectivity between this region and a significant cluster in the thalamus (Figure 3c). The posterior insula seed did not reveal any significant associations. The resulting clusters are displayed in Figure 3a,b and Table 1. 

### 3.2. Correlation between Seed-Based Functional Connectivity and Gold-MSI Subscale Scores

To minimize the number of correlation analyses we performed in our exploratory analysis of the subscales, the effect sizes (i.e., R^2^) were checked for all pairs of correlation between bilateral seeds and clusters in Table 1 and compared between left and right insular seeds. The coefficients of the seed (i.e., either left or right) and cluster correlation with a greater effect size were extracted from CONN. From Figure 4a, the analysis showed that the right dorsal anterior insula was driving the effect (i.e., having a larger effect size than the left dorsal anterior insula) of musical sophistication on seed-based functional connectivity (β = 0.0027, R^2^ = 0.16), the right ventral anterior insula was driving the effect of musical sophistication on seed-based functional connectivity with the cluster in the right insula, right inferior frontal gyrus and right frontal operculum regions (β = 0.0024, R^2^ = 0.26), and the left ventral anterior insula was driving the effect of musical sophistication on seed-based functional connectivity with the two other clusters in superior frontal gyrus (β = 0.0036, R^2^ = 0.24) and thalamus (β = −0.0039, R^2^ = 0.29). Therefore, four Fisher-transformed z-scores of correlation coefficients were extracted between the seed, which was driving the effect and the corresponding clusters. 

Correlation analyses between the z-scores and five musical subscale scores were performed in order to see which subscales of the Gold-MSI might be driving the effect, with Bonferroni correction (*p* < 0.01) for multiple comparison across five musical subscales. As shown in Figure 4b, Active Engagement and Perceptual Ability subscale scores were both correlated with the functional connectivity between the ventral anterior insula and the right insula cluster, right inferior frontal region, and thalamus. Musical training scale was only correlated with the functional connectivity between the right ventral anterior insula and superior frontal region. The Emotion subscale did not correlate with any insula-based functional connectivity. The Singing Ability score was correlated with all insula-based functional connectivity and showed the greatest effect sizes out of all subscales on each pair of correlations between music scores and functional connectivity (Figure 4b). Thus, the effect of musical sophistication on insula-based functional connectivity might be driven by the Singing Ability score. To follow up on this, we looked specifically at the insula-based functional connectivity that showed a significant association with the Singing Ability subscale score.

### 3.3. The Association between Singing Ability and Insula Seed Functional Connectivity

The seed-to-voxel follow-up exploratory analysis revealed that higher scores in the Singing Ability subscale were associated with insula seed-based functional connectivity (voxel *p* < 0.001, FWE cluster *p* < 0.05). When seeding in the ventral anterior insula, participants with higher singing ability scores showed greater functional connectivity between the seed ROIs and the right insula, the right supplementary motor cortex, and left superior frontal gyrus (Table 2). No significant association was observed between singing ability and functional connectivity when seeding in the dorsal anterior insula and the posterior insula.

## 4. Discussion

The current research examined the effect of lifespan musical experiences on the resting functional connectivity of the insula among older adults with a varied range of musical experience. Based on the literature reviewed in the introduction, the insula was divided into three pairs of ROIs: dorsal anterior, ventral anterior, and posterior insula. Results showed a functional distinction between dorsal vs. ventral anterior insula seed regions in their respective associations with musical sophistication. Older adults who had higher Gold-MSI scores showed greater functional connectivity from the dorsal anterior insula to the precentral and postcentral gyri, and from the ventral anterior insula to the right insula, right inferior frontal gyrus (IFG), right frontal operculum, bilateral superior frontal gyrus, and bilateral thalamus. Thus, musical sophistication showed differential patterns of associations with dorsal and ventral anterior insula-based functional connectivity. 

The findings of different patterns of associations between dorsal and ventral anterior insula may support different clusters of insular functional connectivity as identified in previous literature [26]; furthermore, they pointed to distinct patterns of interaction with lifestyle activities such as musical training. While results from Deen et al. (2011) showed that both the dorsal and ventral anterior insula were heavily connected to midline structures such as the cingulate cortex [26], here we see that their interaction with musical experience reveals more of a specific pattern of functional connectivity between the motor and sensorimotor cortices and the bilateral dorsal anterior insula, whereas the ventral anterior insula was more specifically associated with right IFG and inversely associated with subcortical structures that had not been previously reported. Specific follow-up analyses on subscales scores of the Gold-MSI suggested that the functional connectivity between the bilateral ventral anterior insula and cognitive and motor-planning-related regions (i.e., the supplementary motor cortex and superior frontal gyrus) were driven by self-reported singing ability. In this way, musical experiences may offer a window into these different functional connectivity patterns and how they may differentially be affected by lifestyle activities in old age.

The correlation between musical subscale scores and extracted connectivity coefficients suggested that while singing ability was the subscale that specifically interacted with seed-based connectivity from the ventral anterior insula, the effect of music on insula-based functional connectivity was not limited to a single aspect of music, but across a number of facets of musical experience including active engagement, musical training, and singing ability, and to a lesser extent on perceptual ability. These findings are broadly consistent with past studies that have found some benefits of music on perception, cognition, and mental health [23], as well as grey matter volume [36], in older adults. The absence of effect of emotional response to music on insula-based functional connectivity might be because we observed little variability on this subscale score (Figure 1). 

The current study found an association between musical experience and connectivity between the right dorsal anterior insula and superior part of precentral gyrus, which is in the hand motor area [41], and the postcentral gyrus, which serves somatosensory function [42]. However, no effect of musical experience was found on connectivity between posterior insula and sensorimotor regions. It might seem to be unintuitive when considering the findings that the posterior insula was specific to the sensorimotor network while the dorsal anterior insula was associated with the cognitive control network [21,26]. However, the anterior insula also plays a role in sensorimotor function. The anterior insula is a key hub in the bottom-up and top-down regulation between sensory information input and cognitive control networks according to Menon and Uddin’s model [23]. An event-related potentials (ERP) study also found prefrontal N1 and P1 components originating from the anterior insula were associated with integrating visual inputs and motor action [43]. Moreover, the anterior insula was also involved in the inhibition of impulsive motor behavior [44]. Therefore, rather than representation of primary sensorimotor information, anterior insula could also be involved in the interaction between sensorimotor systems and higher-order cognitive systems (e.g., cognitive control), which may benefit from musical experience.

A relationship between musical experience was also found for the functional connectivity between the right ventral anterior insula and a cluster in the right anterior insular region, right inferior frontal, and nearby right frontal operculum. This is in accordance with the functional connectivity network of the ventral anterior insula in the original cluster analysis which defined the three insula regions of interest [26]. The right inferior frontal gyrus has been suggested to support cognitive control processing, such as switching attention between different task demands and inhibiting a prepotent response [45,46]. The right anterior insula, inferior frontal gyrus, and frontal operculum were also found to support detection and responses to salient stimuli, and adaption across different task demands through the ventral attention system [47,48]. Additionally, the inferior frontal cortex, including the inferior frontal gyrus, anterior insula, and frontal operculum, were involved when detecting dissonant stimuli during music listening [49]. Taken together, musical experience may modulate the tightly coupled network of regions spanning the anterior insula to the inferior frontal cortex, and therefore strengthen control processing in older adults.

The bilateral ventral anterior insula showed musical-experience-related increased functional connectivity with the superior frontal gyrus but decreased functional connectivity with posterior thalamus, while the left ventral anterior insula had a larger effect size than the right. The posterior thalamus cluster is close to the medial geniculate body (MGB), which is part of the subcortical auditory pathway [50,51]. Resting-state functional connectivity in the posterior thalamus is also associated with the primary auditory cortex [52], suggesting its role in auditory processing. The potential effect of music on the auditory system is also consistent with longitudinal evidence from an intervention study [53]. Meanwhile, the superior frontal gyrus is involved in executive control function [54,55]. Specifically, the cluster is in the anterior part of the superior frontal gyrus, which is connected to the cognitive control network while the posterior part of the cluster is more connected to sensorimotor regions [56]. Cortical regions in the frontal lobe generally show a greater rate of decline relative to the general rate of cortical decline in aging [57,58]. To combine our observation of a negative association between musical sophistication and functional connectivity in the thalamus, and a positive association between musical sophistication and functional connectivity in the superior frontal gyrus, one possibility is that people with more musical experience might rely more on a highly trained and well-integrated cortical network, including a network of insula and frontal lobe functions, while people with less musical experience might need to compensate by recruiting more activity from the thalamus, thus resulting in greater thalamus connectivity to the ventral anterior insula in less musically experienced individuals. 

The effect of music on ventral and dorsal insula functional connectivity seemed to have a lateralized effect. Some evidence indicated right lateralization of the brain for singing activity, as a comparison to left lateralization of speech activity [59,60,61]. Specifically, the right anterior insula was found to be involved in audio–vocal integration during singing tasks while connected to auditory and motor regions [62]. Right anterior insula functional connectivity might support musical activity more and therefore is modulated by musical experience. This is consistent with our findings that musical experience is associated with right dorsal and ventral anterior insula functional connectivity (Figure 4a). However, connectivity between the left ventral insula and superior frontal gyrus and thalamus showed greater effect size than the right ventral anterior insula (Figure 4a). The left lateralization of the anterior insula and superior frontal gyrus was also observed in previous studies [63,64], which indicates a functional differentiation between the left and right anterior insula. It is possible that musical experience is also associated with connectivity that does not support music directly. To further clarify our findings, the lateralization index was calculated for each functional connectivity between the bilateral insula and their music-associated clusters from Table 1, by (Left − Right)/(Left + Right) as an example of left dominance [65]. However, we did not observe any significant correlations between Gold-MSI subscale scores and lateralization of connectivity (Appendix A Figure A1). This suggests that musical experience does not modulate the degree of functional lateralization, and the lateralized findings in the current report might result from functional differentiation between the left and right anterior insula.

We also found an association between self-reported singing ability and functional connectivity between the ventral anterior insula and the right insula, the right supplementary motor cortex, and the left superior frontal gyrus. These follow-up exploratory results partly overlapped with the above findings of the association between musical composite score and the ventral anterior insula seed functional connectivity. This suggested that self-reported singing ability might be driving the effect of music on insula-based functional networks. Significant increases in cerebral blood flow have been observed in the anterior insula and the supplementary motor area while singing, suggesting these regions were actively involved during singing [66]. Singers also showed altered activity in the anterior insula than non-singers during both resting and singing [67]. Our results suggest that lifetime singing experience may have shaped the insula-based function in our sample as well. Additionally, greater activation has been observed in the supplementary motor area and the insula in musicians compared to non-musicians while improvising music [68]. The strengthened functional connectivity between the insula and the supplementary motor area in participants with more singing experience in our sample may suggest a superior ability to represent sound information and planning complicated motor behavior. Besides brain function, singing activity has also shown benefits for cognition and well-being in older adults [69,70,71]. Given its benefits on the brain and behavior, singing might be a promising candidate for designing music-based interventions in the future.

There are some limitations in the current study, and potential directions for future studies. First, the current results were based on a voxel P threshold at a liberal level (*p* = 0.01) but FWE-corrected cluster-level *p* < 0.05. No cluster survived a more stringent level of threshold (*p* = 0.001), but the reported clusters survived the more stringent cluster-based correction, suggesting a fairly small effect size that affected a relatively larger cluster of functional connectivity. Second, although musical experience showed a relationship to brain networks that support sensorimotor function, we do not have the behavioral data to relate network activity to motor ability measures. Future studies should include motor and cognitive tasks to examine the effect of musical experience on behavioral outcomes. Third, as a cross-sectional study, we cannot assert causal relationships between musical experience and patterns of network activation. Musical training interventions will be necessary to determine whether musical experience is related to changes in brain networks and aspects of perceptual, cognitive and motor performance. Finally, the current sample was extracted from a larger database with inactive older adults (i.e., low physical activity engagement) and with a majority of females, which decreased the representativeness of the results. A larger sample size of balanced demographic factors could improve the generalizability of our findings in future research.

## 5. Conclusions

In conclusion, the current analysis suggests that musical experience, especially singing ability, modulates insula-based functional connectivity, involving regions related to sensorimotor function and cognitive control in older adults. The relationship of musical experience to functional connectivity in the insula-based network extends previous results comparing musicians and non-musicians [31] to a continuum of musical experience in an older adult population. The current findings refine our understanding of the role of musical experiences on insula-based networks in the aging brain. This understanding may serve as a target for future development of music-based interventions for healthy aging. 

## Figures and Tables

**Figure 1 brainsci-12-01577-f001:**
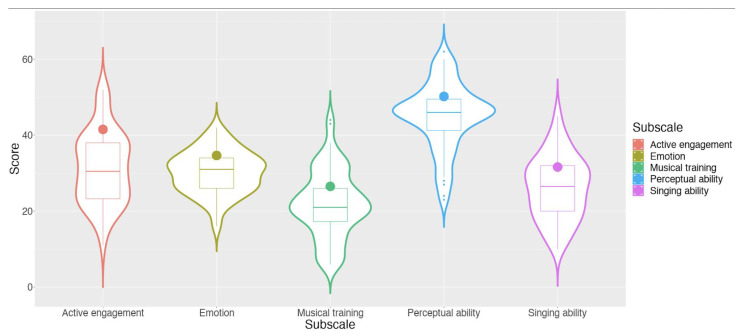
Musical score distribution of current sample compared against normed scores. Solid dots in each violin plot are the median scores from the normed data from Müllensiefen et al., 2014.

**Figure 2 brainsci-12-01577-f002:**
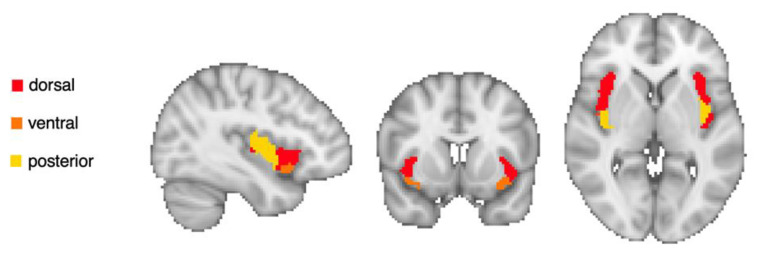
Region of interest (seeds) in bilateral dorsal (red), ventral anterior (orange) and posterior (yellow) insula.

**Figure 3 brainsci-12-01577-f003:**
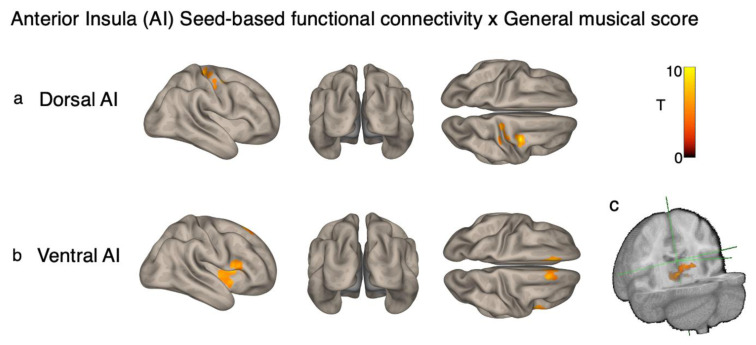
Seed-based connectivity associations with Gold-MSI score. (**a**) Cortical results from dorsal anterior insula seed. (**b**) Cortical results from ventral anterior insula seed. (**c**) Subcortical results from ventral anterior insula seed.

**Figure 4 brainsci-12-01577-f004:**
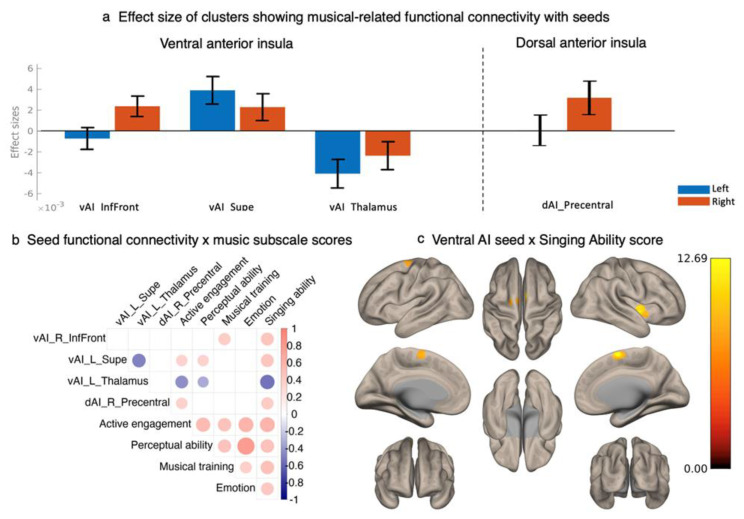
(**a**) Effect sizes of four correlation pairs between seeds and clusters from Table 1. (**b**) Correlation table between seed functional connectivity and music subscale scores. In (**a**) and (**b**), vAI_InfFront refers to the functional connectivity between bilateral ventral anterior insula seed and right insula cluster, right inferior frontal region and frontal operculum, and vAI_R_InfFront refers to the connectivity result from the right anterior insula; vAI_Supe refers to the functional connectivity between bilateral ventral anterior insula seed and bilateral superior frontal gyrus cluster, and vAI_L_Supe refers to the connectivity result from the left ventral anterior insula; vAI_Thalamus refers to the functional connectivity between bilateral ventral anterior insula seed and bilateral thalamus cluster, and vAI_L_Thalamus refers to the connectivity result from the left ventral anterior insula; dAI_Precentral refers to the functional connectivity between bilateral dorsal anterior insula seed and precentral and postcentral gyrus cluster, and dAI_R_Precentral refers to the connectivity result from the right dorsal anterior insula. In (**b**), dots show the significant correlation pairs. Positive correlations are shown in red and negative correlations are shown in blue. Size of dots corresponds to effect size. (**c**) Seed-based functional connectivity from ventral anterior insula associated with score on singing ability.

**Table 1 brainsci-12-01577-t001:** Insula seed functional connectivity clusters associated with musical score.

Seed	Cluster Size (Voxel)	Peak Coordinates	Regions
Bilateral dorsal anterior insula	284	32 −6 48	Right precentral gyrus, right postcentral gyrus
Bilateral ventral anterior insula	588	32 6 12	Right insula, right inferior frontal gyrus, right frontal operculum cortex, right central opercular cortex, right putamen
	287	−6 30 52	Bilateral superior frontal gyrus
	279	−14 −36 4	Bilateral thalamus

**Table 2 brainsci-12-01577-t002:** Insula seed functional connectivity clusters associated with singing ability score.

Seed	Cluster Size (Voxel)	Peak Coordinates	Regions
Ventral anterior insula	126	42 −2 2	Right insula cortex, right planum polare, right central opercular cortex
	89	6 2 62	Right supplementary motor cortex
	82	−18 −2 64	Left superior frontal gyrus, left supplementary motor cortex, left precentral gyrus

## Data Availability

The data in this study are available on request from Arthur F. Kramer and Edward McAuley.

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
