# Peer review of "Musical Experience Relates to Insula-Based Functional Connectivity in Older Adults"

_brainsci, 2022, doi:10.3390/brainsci12111577_

Round 1

Reviewer 1 Report

The authors studied the effect of the musical experience on the Insula's brain network. The method is clear and well described, and the conclusions are in agreement with the results as well as interesting for a possible clinical use. Only minor changes to the text and a small improvement of the analysis are needed.

1) In some parts of the article some references are written in a different form: in most cases they are indicated by square brackets and in some cases with the names of the authors. It may be appreciated to standardize references equally throughout the text.

2) Table 1 and 2 are a bit confused, a clear differentiation between the clusters could be inserted.

3) A suggestion to further improve the analysis: since the results in Figure 4 show a lateralization effect between left and right Insula: a) dorsal anterior --> for the right; b) ventral anterior --> right (cluster 1) and left (cluster 2 and 3). It would be interesting to improve the statistical model to differentiate the brain network related to each side. In this regards, a contrast left>right (or viceversa) can be used in the correlation values between the functional connectivity (seed-to-voxels) and the behavioral data, using the general (Gold-MSI), or, at least, the singing ability score only.

4) In the same way, in the discussion section, a possible interpretation of the different left/right effect of the Insular cortex on the musical experience can be inserted.

Author Response

Response to point 1)

Thank you for the comments. We have fixed the citing errors in the text.

Response to point 2) 

Thank you for this observation. I changed the format of these tables to make it clearer by adding extra lines.

Response to point 3)

Thank you for this suggestion. It is a great point to examine the lateralization and absolutely worth discussing in the report. We calculated the lateralization index for each connectivity by: connectivity Left – connectivity Right)/(connectivity Left + connectivity Right (Seghier, 2007), and correlated the indexes and scores of five Gold-MSI domains with Holm multiple comparison (Holm, 1979). However, We did not observe any significant correlation between either Left or Right dominance of these connectivities and Gold-MSI subscale scores. This indicates that the effect of musical experience potentially has lateralized effects on insula-based connectivity because of functional differentiation, but the degree of lateralization is not associated with musical experience in this context.

Response to point 4)

Thank you for this suggestion. Although there was no correlation between music scores and lateralization index for the connectivity, We added a paragraph in the discussion about the lateralized effect we observed in the current analysis. Potential interpretation based on the functional difference between left and right anterior insula, and lateralization in music activity was added to the discussion part. The correlation analysis of lateralization index was attached as appendix.

Reviewer 2 Report

I read the paper by Meishan Ai et al. and I think (for the part I can judge better, the intro and the behavioural part) it is suitable for publication with relatively few changes.

The main points I rise are three:

1. there is only a small selection of papers in the introduction, and because of this, the introduction reports only one side of the story (incomplete, in my opinion)

2. It's a pity (but authors acknowledge this in the discussion) that only the GOLD-MSI was used in the study and no other behavioural measure was taken. I wonder whether this was just a subset of a larger study and whether the authors are reporting here only the part related to music. Unfortunately, no much can be done on this at this stage. In particular, the GOLD-MSI (although highly used in the field) it is simply a self report questionnaire. I was wandering, if there is any pre-registration of this study. This would help to understand whether the study is part of something larger and the particular choice of this tool only (GOLD-MSI).

3. Why authors do not share their data and digital materials?

About point #1. If I had to synthesise the intro, the emerging motto is "music is good for you". But actually, the story is a bit more complex than that and there is a large discussion about. This discussion (and the various nuances) should be acknowledged in the introduction. In general, music ability has a (short) transfer on sensory ability. And sensory ability (in turn) is related to better cognition (long transfer). Auditory abilities seem connected to cognition (eg processing speed, Grassi & Borella, 2013) and music activity across the lifespan keeps central auditory abilities of musically sophisticated individuals better than those without musical sophistication (Zendel & Alain, 2012). However, in older musicians and nonmusicians, we found no association between sensory ability and cognition  (Grassi & at., 2017) and although musicians had overall better sensory AND cognitive performances. Accoding to recent meta-analysis, music training seem to have little effects on cognition (Sala and Gobet, 2017). In brief, we are still exploring. I would also stress that none of the above results suggests a cause-effect of music training on anything (see Schellenberg, 2020). I also suggest authors a comprehensive review that includes more on this topic (Swaminathan & Schellenberg, 2019).  

I hope this helps

References

Grassi, M., & Borella, E. (2013). The role of auditory abilities in basic mechanisms of cognition in older adults. Frontiers in Aging Neuroscience, 5, 59.

Grassi, M., Meneghetti, C., Toffalini, E., & Borella, E. (2017). Auditory and cognitive performance in elderly musicians and nonmusicians. PLoS One, 12(11), e0187881.

Sala, G., & Gobet, F. (2017). When the music's over. Does music skill transfer to children's and young adolescents' cognitive and academic skills? A meta-analysis. Educational Research Review, 20, 55-67.

Schellenberg, E. G. (2020). Correlation= causation? Music training, psychology, and neuroscience. Psychology of Aesthetics, Creativity, and the Arts, 14(4), 475.

Swaminathan, S., & Schellenberg, E. G. (2019). Music training and cognitive abilities: Associations, causes, and consequences. The Oxford handbook of music and the brain, 644-670.

Zendel, B. R., & Alain, C. (2012). Musicians experience less age-related decline in central auditory processing. Psychology and aging, 27(2), 410.

Author Response

Response to point 1)

Thank you for pointing out the limited scope of the introduction, and for providing all the sources of studies. We agree that the modulation of musical experience on cognitive and brain health is complicated and more than just having a benefit. We have cited these studies and some others and edited the first part of the introduction, in order to strengthen the background information. Meanwhile, we did keep the majority of the introduction focused on insula-related evidence and its association with musical experience, because we are looking at insula-based functional connectivity specifically.

Response to point 2)

Thank you for bringing up this point. The current sample is a subset of a larger clinical trial of exercise effect on cognition and brain health (https://clinicaltrials.gov/ct2/show/NCT01472744). All the imaging data were taken from the baseline assessment of this clinical trial. There is no pre-registration for this study. We mailed the GOLD-MSI questionnaire to the participants as well as consent after they completed the trial, and they finished the questionnaire at home. Indeed, this is the only music-related measure, and not the primary measure of this clinical trial. This information is provided in lines 160-168. We choose GOLD-MSI questionnaire because this the most economical and effective way to get music-related information from the participants, compared to other in-person or electronic device required assessment.

Response to point 3)

Thanks for the question. The data and materials could be shared upon request from Dr. Arthur Kramer and Dr. Edward McAuley. I have specified this under the Data Availability Statement.

Reviewer 3 Report

This is an excellent study and clearly presented manuscript. I have only a few minor suggestions:

Introduction

Line 44 – To clarify that the authors are focusing on music-making not music listening, recommend adding a parenthetical definition to ‘musical activity’ – e.g. ‘singing, playing an instrument’.

Lines 57-58 – several repeated words/phrases in this sentence; recommend revising for clarity

Line 71 – recommend changing the parenthetical reference to Craig, 2003; 2011 to a numerical citation for consistency

Lines 78, 87 & 320 – please add numerical references corresponding to Deen et al. (2011) and Lordier et al. (2019) 

Results

Tables 1 & 2 – these tables are quite difficult to read; recommend adding shading/spacing/additional lines to clarify which regions relate to each seed/cluster size/peak coordinates

Lines 376 – 379 – discussion of the results of the recent intervention study demonstrating effects of piano training in older adults on cortical thickness of auditory structures (Worschech, Florian, et al. "Evidence of cortical thickness increases in bilateral auditory brain structures following piano learning in older adults." Annals of the New York Academy of Sciences (2022).) would be a valuable addition here  

Author Response

Because that there are multiple comments, I cited all of them below for being easier to keep track of each point.

Comment: Line 44 – To clarify that the authors are focusing on music-making not music listening, recommend adding a parenthetical definition to ‘musical activity’ – e.g. ‘singing, playing an instrument’.

Response: Thank you for the suggestion, we have rephrased that sentence with specifying musical activity.

Comment: Lines 57-58 – several repeated words/phrases in this sentence; recommend revising for clarity

 Response: Thank you for pointing this out. We have rephrased this sentence to make it clearer.

Comment: Line 71 – recommend changing the parenthetical reference to Craig, 2003; 2011 to a numerical citation for consistency

Response:  Thanks for this observation. We have fixed the citing formatting error in the text.

Comment: Lines 78, 87 & 320 – please add numerical references corresponding to Deen et al. (2011) and Lordier et al. (2019) 

Response: Thank you for the observation. We have added numerical references for these papers.

Comment: Tables 1 & 2 – these tables are quite difficult to read; recommend adding shading/spacing/additional lines to clarify which regions relate to each seed/cluster size/peak coordinates

 Response: Thank you for the suggestions. We have added extra lines to reformat the table and make it clearer.

 Comment: Lines 376 – 379 – discussion of the results of the recent intervention study demonstrating effects of piano training in older adults on cortical thickness of auditory structures (Worschech, Florian, et al. "Evidence of cortical thickness increases in bilateral auditory brain structures following piano learning in older adults." Annals of the New York Academy of Sciences (2022).) would be a valuable addition here  

Response: Thank you for providing information about this paper. This is great longitudinal evidence to cite about musical effect on brain. We have added it in the discussion.